An optimized BERT fine-tuned model using an artificial bee colony algorithm for automatic essay score prediction

Chassab Ridha Hussein
Zakaria Lailatul Qadri lailatul.qadri@ukm.edu.my
Tiun Sabrina
Faculty of Information Science and Technology, Universiti Kebangsaan Malaysia , Bangi, Selangor , Malaysia
Alatas Bilal
Electronic publication date: 2024 Sep 24
Publication date: 2024
Volume: 10
Electronic Location ID: e2191
Received 2023 Feb 9; Accepted 2024 Jun 21
Copyright: © 2024 Chassab et al.
Copyright year: 2024
Copyright holder: Chassab et al.
License: This is an open access article distributed under the terms of the Creative Commons Attribution License, which permits unrestricted use, distribution, reproduction and adaptation in any medium and for any purpose provided that it is properly attributed. For attribution, the original author(s), title, publication source (PeerJ Computer Science) and either DOI or URL of the article must be cited.
License URL: https://creativecommons.org/licenses/by/4.0/

Keywords: Automatic essay score, Catastrophic forgetting, Prediction error, Bi-directional encoder representation from transformation, Artificial bee colony algorithm, Freezing mechanism

Funding: Faculty of Information Science and Technology, Universiti Kebangsaan Malaysia This research was supported by the Faculty of Information Science and Technology, Universiti Kebangsaan Malaysia. The funders had no role in study design, data collection and analysis, decision to publish, or preparation of the manuscript.

==============================
Background

The Automatic Essay Score (AES) prediction system is essential in education applications. The AES system uses various textural and grammatical features to investigate the exact score value for AES. The derived features are processed by various linear regressions and classifiers that require the learning pattern to improve the overall score.

Issues

Moreover, the classifiers face catastrophic forgetting problems, which maximizes computation complexity and reduce prediction accuracy. The forgetting problem can be resolved using the freezing mechanism; however, the mechanism can cause prediction errors.

Method

Therefore, this research proposes an optimized Bi-directional Encoder Representation from Transformation (BERT) by applying the Artificial Bee Colony algorithm (ABC) and Fine-Tuned Model (ABC-BERT-FTM) to solve the forgetting problem, which leads to higher prediction accuracy. Therefore, the ABC algorithm reduces the forgetting problem by selecting optimized network parameters.

Results

Two AES datasets, ASAP and ETS, were used to evaluate the performance of the optimized BERT of the AES system, and a high accuracy of up to 98.5% was achieved. Thus, based on the result, we can conclude that optimizing the BERT with a suitable meta-heuristic algorithm, such as the ABC algorithm, can resolve the forgetting problem, eventually increasing the AES system’s prediction accuracy.

Introduction

Assessment is one of the main components of evaluating student abilities in education systems. Technology development significantly impacts various learning platforms’ E-learning systems (Almaiah & Alyoussef, 2019; Tawafak, Romli & Alsinani, 2019). The tremendous progress of computerized systems contributes highly to online assessment systems. Most systems only focus on multiple-choice questions to assess student learning and memory ability. The multiple choice-based assessment procedures are factual and exact answers (Alshammari, 2020). However, assessment systems face several difficulties when investigating subjective solutions. Therefore, researchers use the Automatic Essay Scoring (AES) system to maximize online assessment efficiency (Farag, Yannakoudakis & Briscoe, 2018; Ramesh & Sanampudi, 2022). The AES system should focus on subjective answers because they may have grammatical errors, spelling errors, different essay structures, paragraphs, words, and sentences (Das et al., 2022; Abdulraheem, Zakaria & Omar, 2021). Therefore, the AES system should be created by considering these essay features. The AES process uses predefined programs to allot the score or grade to the student’s written document. During this process, natural language processing (NLP) is incorporated with the AES system to explore the textual entities and discrete categories (Pashev, Gaftandzhieva & Hopteriev, 2021; Rahayu & Sugiarto, 2020). Several AES systems give inappropriate results because of improper assessment of student answers. Hence, the AES training system is developed by considering the hand-score process. The training set considered text surface features, subordinate clues, lower-case letters, word count, sentence count, and upper-case letters. These features construct a mathematical model using different machine-learning techniques (Mayfield & Black, 2020; Wang et al., 2018; Cozma, Butnaru & Ionescu, 2018; Chassab, Zakaria & Tiun, 2021a). This AES model predicts the score value for the given input. The AES model supports various features such as argument location, prompt topics, cohesion, author’s agreement level, texture features, and argument error (Deane et al., 2021). The constructed AES models are more similar to the manual scoring process, and the extracted features are investigated by applying the linear regression technique. In addition, various machine learning techniques (Sarker, 2021; Naswir, Zakaria & Saad, 2022), such as support vector machine (SVM), regression models, k-nearest neighboring (KNN), and neural models (NN), highly process the extracted features (Ratna et al., 2019). These machine-learning techniques are categorized into supervised and unsupervised learning processes (Martinc, Pollak & Robnik-Šikonja, 2021; Bhutto et al., 2020; Kaggle, 2012). In supervised learning, generated training patterns are used to allocate the score values for user input, and unsupervised learning allocates the score value without having any learning patterns. Optimization techniques are incorporated into this method to improve overall efficiency. However, the existing systems consume high computation difficulties such as training time, weight parameter update, etc.

The deep learning (DL) approach is highly utilized in several applications to perform prediction and classification (Chassab, Zakaria & Tiun, 2021b; Abdulraheem, Zakaria & Omar, 2021; Hamzah, Mohd & Zakaria, 2022). The DL has several sub-neural models such as convolution neural networks (CNN), long short-term memory (LSTM), recurrent neural networks (RNN), sequence neural networks, deep belief networks (DBNs), generative adversarial networks (GANs), and radial basis neural networks (RBN). These DL approaches utilize different learning functions to predict the output value. However, these DL methods fail to address the catastrophic forgetting problem while training and classifying data. This research problem causes a reduction in the score detection efficiency and creates high computation complexity. Thus, this study’s research issue is overcome by incorporating the BERT model with the neural model. Here, the BERT model extracts the embedding vectors from the input text. Then, the models are investigated by applying the multi-head attention model, which predicts a similar pattern. Afterward, the unfreezing mechanism-related issues are solved by fine-tuning the neural parameter with the help of the Artificial Bee Colony (ABC) algorithm. The ABC algorithm selects the optimized features using the bee food searching process. This process improves the overall score prediction accuracy and minimizes output deviations. The main reasons for choosing the ABC algorithm are the effective selection of features and minimum error rates. The created AES system based on the optimized BERT model uses the two datasets for evaluation. In addition, the optimized BERT-based AES system is assessed using accuracy, F1-score, and Quadradic Weighted Kappa (QWK) metrics.

The rest of the article is arranged as follows: “Related Work” discusses the different researchers’ opinions about AES prediction. “ABC-BERT-FTM Model” analyzes the working process of the optimized BERT model based on AES prediction and efficiency in “Results”. Finally, the conclusion is given in “Discussion”.

Related work

This section describes how the different researchers work on the AES prediction system. This research analysis was used to gain knowledge about the score prediction techniques.

Zhang & Litman (2019) created an AES detection system using the co-attention deep learning (CA-DL) architecture. The network used the convolution layers that derive the embedding word vectors from the Glove pre-training model. The results were analyzed by applying the long short-term memory neural network that creates the sentence embedding from the student’s answer. A co-attention layer was applied to obtain a similar sentence between the model and the student’s answer. Finally, the last CNN layer predicts every student’s answer score. This process used the Aligned Scores and Performances (ASAP), in which the system achieved 81.5% accuracy.

Beseiso & Alzahrani (2020) utilized similarity measures to address the similarity of the answers’ embedding vectors. Since the door has been opened for encoding additional features thus, other features such as embedding features and word2Vec features engineered were still needed, especially for the similarity features where it might be suitable for encoding its resulted values rather than the simple pairwise matching between the embedding vectors addressed by Beseiso & Alzahrani (2020). Incorporating these features into the embedding would benefit both supervised and unsupervised learning.

Hendre et al. (2020) have compared several embedding approaches for the AES. The authors used the ASAP benchmark dataset. Conventional vector representations such as TFIDF and Jaccard have also been used. Different embedding techniques, such as Glove, Elmo, and Google Sentence Encoder (GSE), were also utilized. Finally, the authors used cosine similarity to find the similarity between student response vectors and instructor replies. The results indicate that GSE had the strongest correlation.

Ridley et al. (2021) applied multi-task learning (MTL) and a deep learning model (DL) to perform the AES task. The student essay was evaluated based on traits instead of the holistic process. Various structural features such as topic discourse, essay vocabulary, size of vocabulary, and essay organization are extracted from the answers. Then, word embedding details were derived using the convolution layer, which utilizes the Glove pre-training model. The obtained embedding features were processed using the short-term long memory (LSTM) network that allocates the score value for every answer. Here, Aligned Scores and Performances (ASAP) dataset information was processed with the help of MTL and the pooling attention layer. Parts of Speech embeddings were used to extract the features during the analysis. Then, the system ensures 76.4% accuracy compared to the standard classifiers.

Li, Chen & Nie (2020) utilized two deep learning models to perform the AES task: LSTM and CNN. The convolution structure was initially utilized to derive the word vector from the student’s answer. The vectors were derived according to the help of the Glove pre-trained model. The extracted word vectors were given input to the LSTM network that predicts the score value using the pre-trained model. This process used ASAP, and the system ensures 72.65% accuracy.

Rodriguez, Jafari & Ormerod (2019) interested in developing the (AES) to reduce the difficulties in manual essay scoring. However, the AES system should consider prompt content, cohesion, development ideas, and coherence. These parameters were difficult to address by using the existing AES techniques.

Gaheen, ElEraky & Ewees (2021) implemented an Elitist-Jaya optimized algorithm based on the neural network to predict the essay score for Arabic students. The research used 240 students’ essay information, and the collected details are investigated according to the network layer. The network transfers the answer text into the digit matrix. The generated matrix was analyzed by comparing it with the trained dataset. During the analysis, the Jaya optimization technique updates the network parameters, which improves the recognition rate.

Rajagede (2021) increased the performance of the essay scoring system by applying a Tree-structured Parzen Estimator-based neural model. This work used the richer features derived from the BERT model that the neural model processes. The extracted features were processed using the XGBoosting technique, which generates the pre-training model. The pre-trained model helped recognize the essay score with 0.829% accuracy on the Ukara dataset. Researchers used different neural models and embedding systems to improve the score prediction rate.

Ma, Cheng & Shi (2020) introduced brain storm optimization algorithm to improve learning efficiency by creating the orthogonal learning design. This process used quantitative association rules to address comprehensibility, confidence, and support problems effectively. Ma et al. (2021) developed copper-burdening systems by applying many-objective optimization algorithms. The system used the reinforcement learning algorithm to adapt the copper burdening with defined constraints. The generated learning rules are used to address the copper-burdening problems effectively.

Yu et al. (2021) applied multi-objective differential evolution methods to address the optimization problems. This study used 56 objective problems to explore the system objectives effectively.

However, the existing systems fail to address the catastrophic forgetting problem and prediction error. These concerns affect the score prediction efficiency. The research issues are overcome by applying the unfreezing-based BERT model fine-tuning process. Then the main objective of the work is listed as follows: to minimize the catastrophic forgetting problem while updating the network parameters is done by applying the optimized unfreezing mechanism,

to minimize the prediction error rate, select the network parameter using the ant bees searching-based parameter selection,

to maximize the accuracy of the essay score prediction by incorporating the unfreezing mechanism with the BERT model.

Abc-bert-ftm model

This system aims to reduce the difficulties in the Bidirectional Encoder Representation from Transform (BERT) fine-tuning process. The BERT model can process forgetting contextual information, which requires an effective network parameter to improve the score prediction process. Existing research techniques fail to utilize the unfreezing mechanism to optimize the network parameters. The fine-tuning procedure updates the network parameters, such as weight and bias values. These values are more valuable for minimizing the prediction error rate and maximizing the precision value. The precision value is determined based on the Quadradic Weighted Kappa (QWK) range concordance value. If the essay has a high QWK value, it achieves a high prediction rate. This system uses the ASAP dataset (Kaggle, 2012) and the Educational Testing Services (ETS) corpus dataset (Linguistic Data Consortium, 2014) to investigate the system’s effectiveness. Most of the researchers utilized these datasets for their research purposes. In addition, the large volume of information in the dataset helps to identify the system’s effectiveness. These datasets have been divided into training (70%) and testing (30%). The training datasets extract the features and training patterns to predict the score values. The working process of the optimized BERT model is illustrated in Fig. 1.

Figure 1 ABC-BERT-FTM model.

Figure 1 demonstrates the working process of the automatic essay score prediction system. Here, the student answers are given as input to the system. The inputs are analyzed by applying pre-processing techniques for removing the punctuation, Uniform Resources Locator (URLs), lower casing, tokenization, stemming, and lemmatization process. The pre-processing technique simplifies the computation difficulties and maximizes score prediction accuracy. Then, the BERT embedding model is applied to analyze the pre-processed input to derive the score values for the given input. The obtained output values are compared with the training patterns to minimize deviations between the actual and computed output values. The ABC algorithm is applied to fine-tune the network parameter, minimizing the output deviations and prediction error rate according to the comparison values. The detailed explanation of the score prediction system is described as follows: A) Phase 1: Pre-processing

The first step in this work is to remove unwanted information from students’ answers. The pre-processing procedure helps to improve the overall scoring rate and reduce computation complexity. This step eliminates unwanted characters, spaces, words, and other details during the assessment process. The noise removal process applies natural language processing (NLP) and an embedding model. The noise removal process is illustrated in Fig. 2.

Figure 2 Working process of corpus noise removal.

The collected information is investigated to remove the punctuation marks, stop words, and URLs. After that, lowercasing, lemmatization, stemming, and tokenization were performed to improve further analysis. The main intention of lemmatization and stemming is to minimize the inflectional forms and generalize words. For example, if the sentence “organize, organizing and organizes” is generalized as “organize.” The stemming process cuts the word’s end and creates a general word. Generally, stemming is a way of removing catastrophic derivational affixes. Lemmatization is analyzing words according to the vocabulary and a morphological way to remove inflectional endings. This process follows a few rules while performing the stemming and lemmatization process. Words are changed according to the rules “SSES→SS; IES→I; SS→SS; S→remove S.” Depending on the rules, the following examples are given “caresses→caress; Ponies→poni; caress→caress; cats→cat”. After that, tokenization is performed in which the strings are split into the number of tokens. The texts are divided into sentences, sentences are changed into words, and respective regular expressions are extracted. This process is repeated continuously, and noise information has been eliminated from the corpus details. Then the noise-removed details are fed into the word embedding model to extract the student answers’ patterns. B) Phase 2: Word embedding analysis for a score prediction

The noise-removed information is fed into the embedding model to investigate the text. The embedding process generates a vector for every word, recognizing similar document words. The word embedding mapping process is denoted as φ, which is defined as W→Rn; word space is represented as W, and the vector space dimensional is defined as Rn. The φ process utilizes the Bidirectional Encoder Representation from Transform (BERT) model. The embedding model has an encoder transformer and applies Masked Language Modelling (MLM) to mask words according to their position information. Then, mixed information is incorporated into the learning process to maximize score prediction accuracy. The embedding model uses the transfer learning model and fine-tuning parameters to minimize the convergence time. The BERT model structure is illustrated in Fig. 3.

Figure 3 BERT model structure.

The BERT model has 24 transform blocks (L), 1,024 hidden layer sizes (H), and 16 attention heads (A) for processing the inputs. The BERT model has different training elements such as CLS (first token used to perform the classification process and works with the SoftMax layer), SEP (sequence delimiter token used to train the sequence of words; if the sentence has a single sentence, it is included at the end) and MASK (applied during the pre-training to mask words). Figure 3 illustrates the sentence, “My cat is cute; he likes playing.” The unique token and each vector word denote the input layer. Then Out-of-Vocabulary (OOV) words are removed from the inputs. Token embedding is performed by giving the token Id; sentence embedding is performed by computing the differences between sentences A and B. C) Phase 3: Optimized BERT model using ABC algorithm:

Finally, transformer positional embedding is applied to compute the word sentence. Then, grammar errors are computed from sentences and words in the pre-training process. Here, the Language_check package is applied to eliminate grammatical errors. Pre-processing concerns are also performed to improve the accuracy of the overall score prediction. The output of position embeddings is fed into the BERT transformer model to identify the exact output, and the transformer model structure is shown in Fig. 4.

Figure 4 BERT transform model working process.

Figure 4 illustrates the transform model working process, which includes masked multi-head and multi-head attention neural models used to predict the score values for student answers. During the computation, the network consists of several parameters optimized in every iteration to minimize prediction error. The high dimensionality of network features can lead to over-fitting and convergence issues. Therefore, network parameters require fine-tuning to maximize the overall prediction rate. The BERT model uses a feed-forward neural model to compute the output value. The network has three layers: input, output, and hidden; each layer has a dynamic association with the NLP attention layer. The attention layer uses both directions while reading the answers (right to left to right), maximizing the overall score prediction rate. The ABC algorithm process is illustrated in Table 1.

Table 1 ABC algorithm.

Step 1: Initialize xij using the maximum and minimum parameters in the search space.
Step 2: Analyzing x using the employee bee in xij
Step 3: compute the fit(xi) in employee bee
Step 4: compare the new and old food sources according to the nectar value.
Step 5: Check the termination condition of the employee bee (search = end of the food source) and move to the onlooker bee.
Step 6: Estimate pi value for the food source.
Step 7: Select the food source according to pi
Step 8: compare the new and old food sources in the onlooker phase
Step 9: Check the termination condition of the onlooker bee (search = end of the food source) and move to the scout bee
Step 10: Compare the objective function and minimize the food source list.
Step 11: Update the solution and select the parameter according to the search process.	

The optimization algorithm-based network updating process minimizes catastrophic forgetting problems and effectively stores all information. During the analysis, the network utilizes its memory to compare new parameters with existing learning patterns, updating learned parameters to predict new incoming related outputs without causing difficulties. This continuous learning procedure reduces the need for a lookup table and maintains the relationship between parameters. Prior knowledge and network parameters are useful in preventing interference forgetting. The system is continuously trained according to selected parameters and data, reducing deviation between actual and computed output values and providing prior knowledge of essays, words, sentences, and phrases to improve overall essay analysis efficiency. Extracted data provides prior knowledge to classify new information with minimum deviation error and high recognition accuracy. The optimized algorithm-based pre-trained model provides specific patterns for analyzing new data with proper activation functions and hidden layer weight parameters, effectively resolving the catastrophic forgetting problem. During the analysis, around 10 to 20 features (ants) are utilized to compute the optimized solution. Then, efficient pheromone evaporation values control the pheromones that evaporate in the search space. This study uses 0.5 to 0.7 as the pheromone’s coefficient component. In addition, 1 to 10 pheromone intensity values are utilized to determine the strong level of pheromones. Then, 0.5 to 1.0 heuristic factors are chosen to estimate the decision value of the features, and finally, 0.1 to 0.3 randomness values are utilized. D) Phase 4: Evaluation of BERT vs. optimized BERT

To examine the score value, the embedding process examines the word’s position and the relationship between the words and sub-words. In addition, the transformer process takes tokens as input, which the intermediate layer processes to get the output value. The main difficulty in text analysis is a limited context learning model, which requires continuous learning and training. Therefore, the BERT model extracts the various information trained by giving the language modeling. The modeling process initiated its training process with new aspects. The training process is performed using the freezing mechanism in which entire architectures are trained using the activation layer, and a few layers are frozen to improve the fine-tuning process. The whole dataset utilized as input and output during the training is estimated using the SoftMax layer. The considered output values are compared with the training pattern, and the deviations are calculated using the loss function. Suppose the system has a few error values, which are reduced by applying the backpropagation procedure. The network parameters are backpropagated to the layers and are updated continuously to minimize the prediction error rate. Initially, few-layer weight values are frozen, and the remaining layer’s network parameters are updated by computing the new weight value. The network requires special attention while handling the unfreezing parameters because the network has constant freeze values. The unfreezing network parameters-related learning rate has been increased gradually in every iteration. The variations in the learning rate affect the score prediction task. Therefore, a layer-wise fine-tuning procedure is applied to update the network parameters. The network uses every layer’s stochastic gradient descent (SGD) learning process. The SGD parameters are updated according to Eq. (1)

(1) θt=θt−1−η.∇θJ(θ).

In Eq. (1), the objective function of the gradient is represented as ∇θJ(θ) and the individual learning rate is signified as η. The η value is updated with several learning rates. {η1,η2,…..ηL} which is performed during the fine-tuning process. In Eq. (1), the learning rate is applicable for the L layer, and the learning rate is applied to the last layer ηL. Similarly, the network has several parameters mentioned as {θ1,θ2,…..θL}. Then the network parameter’s fine-tuning is defined by Eq. (2).

(2) θtl=θt−1l−ηL∇θtJ(θ).

According to Eq. (2), the network parameters are fine-tuned; during this process, the network uses the 2e−5 initial learning rate and lower layer have ηL−1=ηL/1.1. In the network, higher layers have information, and the lower layer consists of much information used to identify the similarity between the answered essay and the user-written essay. The gradual unfreezing mechanism creates the catastrophic forgetting problem. This problem is the neural model completely forgetting the previously learned information, which causes a reduction in the overall network performance. Therefore, unfreezing action is applied on the BERT model’s last layer, and Eqs. (1) and (2) are utilized to fine-tune the network parameter. The residual layers are frozen during training, and this process is repeated continuously to eliminate the catastrophic forgetting problem. The network effectively utilizes the freezing mechanism to solve the prediction issues. However, the network requires optimized parameters while updating the weights and bias values. The ABC algorithm selects the optimized values. The ABC algorithm chooses the parameter based on the bee food analyzing process. The ABC algorithm has two important concepts: artificial bees and food sources. These two sources are widely applied to selecting high-nectar value-related food. The parameters are chosen based on the food source position and nectar quantity used to resolve the optimization problem. The algorithm utilizes the scout, onlooker, and employee bees which are highly utilized in food searching. The search space has various food resources, which are initialized using Eq. (3)

(3) xij=xminj+rand(0.1)(xmaxj−xminj)

In Eq. (3), solutions are analyzed in the ith search space and search space having the BN solutions. The solutions depend on the neural fine-tuning parameters (weight and bias value). The parameters are predicted in the solution’s maximum ( xmaxj) and minimum ( xminj). The employee bee analyzes the solution in ith space and high nectar value related foods chosen. The identified food sources are transferred to the scouts, and the scouts investigate the unidentified food sources. The employee bee searching process is described using Eq. (4)

(4) vij=xij+φij(xij−xkj).

Here, k is the value taken from 1 to BN, a random decimal value φij is from −1 to 1, and the newly searched food is defined as the vij that is computed from xi. At last, the selected sources are transferred to the onlooker bees to predict the optimized solution. Therefore, nectar values are computed, and the maximum nectar value-related solutions are chosen as the optimized network parameter. The new solution is estimated using Eq. (5).

(5) pi=fit(xi)∑n=1BN⁡fit(xn).

The new solution is selected according to Eq. (5), which is used to identify the optimal solutions. After identifying the solutions, the optimal new solutions are obtained by performing the mutual learning that is defined in Eq. (6)

(6) vij= {xij+φij(xKj−xIj),Fiti<FitkxKj+φij(xij−xkj)Fiti≥Fitk.

Equation (6) represents current and neighbouring food sources’ fitness levels as the Fiti and Fitk. A Uniform random number, represented as φij, is a numerical value that ranges from 0 to F. F represents the non-negative constant value or mutual learning factor. The generated new solution-based neural model weight parameters are used to update the network parameter. Effective learning and solution-updating procedures effectively reduce the exploration and exploitation issue. Then, the efficiency of the created system is evaluated using the experimental analysis discussed in “Results”.

Results

This section analyzes the ABC-optimized BERT Fine-tuning Modelling (ABC-BERT-FTM) approach based on automatic essay score prediction efficiency. The system uses the ASAP and ETS corpus datasets to evaluate its effectiveness. The ASAP dataset from the Kaggle.com repository contains information from student questions and answers and multiple teacher assessments related to essay answers. The dataset includes persuasive, narrative, and expository essays from eight grade-level students, with 1,785 training sets and 592 essays for final evaluation, with an average length of essays of 350 words. The system also uses the Educational Testing Services (ETS) corpus dataset, which includes fifth-grade students’ narrative essays. This section analyzes the ABC-optimized BERT Fine-tuning Modelling (ABC-BERT-FTM) approach based on automatic essay score prediction efficiency.

The training dataset consists of 11,000 essays, and the final evaluation set has 1,100. The average length of essays is 407 Rds. The details of the collected essay are divided into training (70%) and testing (30%) to evaluate the system’s efficiency. The collected information is processed by extracting the various embedding vectors. These vectors are more valuable for predicting the score values. The score values are allocated according to the QWK, depicted in Table 2.

Table 2 QWK range concordance.

S. No	QWK range	Concordance	
1	+Ve values	Poor	
2	0.01 to 0.25	Slight	
3	0.26 to 0.50	Fair	
4	0.51 to 0.75	Moderate	
5	0.76 to 0.85	Substantial	
6	0.86 to 0.9	Almost perfect	
7	0.91 to 1	Perfect	

According to Table 2, the scores are allocated to the student’s answers. The system ensures effective results by successfully utilizing the network parameters, activation function, and learning parameters. Then the attained results are compared with the existing methods, such as co-attention deep learning (CA-DL) (Zhang & Litman, 2019), BERT Embedding (Beseiso & Alzahrani, 2020), Cosine similarity and vector modeling (CS-VM) (Hendre et al,. 2020), multi-task learning (MTL) with Deep Learning Model (DL) (MTL-DL) (Ridley et al., 2021), and hybrid deep learning model (HDL) (Li, Chen & Nie, 2020). The scores are predicted based on the training patterns because different patterns are investigated during the training, and the values are allocated based on Table 2. These training patterns improve the accuracy of overall score prediction. The training efficiency is explored using the accuracy, F1-score, and QWK measures estimated using Eqs. (7)–(9).

(7) Accuracy=α+βα+β+γ+θ.

In Eq. (7), α is represented as the true positive value = correctly classified according to the user query. β is denoted as true negative-the model correctly identifies the negative class, γ is denoted as false positive-incorrectly identifying the output for the query, and θ is defined as a false negative that indicates wrongly identifying the specific condition.

(8) F1−score=2α2α+γ+θ

(9) QWK=2∗(α∗β−γ∗θ)(α+γ)∗(γ+β)+(α+θ)∗(θ+β).

The obtained results are more relevant to the student’s exact output score. The deviation between the output values is evaluated using the Mean Square Error (MSE) and Root Mean Square Error (RMSE) values. These metrics are computed using Eqs. (10) and (11).

(10) MSE=1n∑i=1n⁡(yi−yi^)2

(11) RMSE=1n∑i=1n⁡(yi−yi^)2

In Eqs. (10) and (11), n is denoted as the number of samples in the search space, yi is defined as the computed output value and yi^ is defined as the actual output value.

The BERT model uses different layers to analyze the word inputs, which requires fine-tuning. The fine-tuning procedure uses unfreezing to update the network parameters and improve overall performance. During the updating process, the ABC approach selects the weight parameters theta sub t to the l in the search space The ABC algorithm uses the onlooker bees, employees, and scouts bees to search and select the best features based on the fit(xi). The function analyses the entire solution, and high-rank features are selected as the best weight parameter. During this process, the network uses the vij and the learning process selects the best parameter according to Fiti<Fitk and Fiti≥Fitk condition. The effective process of network parameter updating procedure reduces the optimization problem effectively. Then the obtained graphical results are illustrated in Figs. 5 and 6.

Figure 5 Accuracy and F1-score analysis.

Figure 6 Error value and QWK analysis.

Figures 5 and 6 illustrate that the ABC-BERT-FTM approach attains the maximum results while predicting the score value to the student input. Here, the approach uses the NLP techniques that cut and eliminate the irrelevant information in the document described in “ABC-BERT-FTM Model”. The network utilizes the noise removed, lemmatization, and stemming performed inputs, reducing the computation difficulties. In addition, networks have the self-attention model that uses the neural network learning function, learning parameter, batch normalization, and SoftMax activation function to predict the output value. The network uses the ABC-based selected network parameter during the computation, which helps fine-tune the network performance, reducing the deviation between the score values.

Along with the effectiveness of the introduced ABC optimized BERT Fine-tuning Modeling (ABC-BERT- FTM) is compared with the normal BERT-based AES model (Rodriguez, Jafari & Ormerod, 2019), Elitist Jaya optimized neural networks (EJNN) (Gaheen, ElEraky & Ewees, 2021). Tree-structured Parzen Estimator-based Neural Model (TSPE-NN) (Rajagede, 2021). Among the different approaches, the main reason for selecting this optimization algorithm is the effective utilization of the algorithmic process, function, and score analysis process. Although these methods work perfectly, the AES system’s effectiveness should be improved. Therefore, the ABC algorithm is incorporated with the BERT model to improve the overall AES system performance. Then the obtained resultant graphical representation is illustrated in Fig. 6.

Figure 7 clearly shows that the introduced ABC-BERT-FTM approach attains a high accuracy (98.51%) value compared to the other optimized classifiers such as BERT-AES(94.56%), EJNN (95.99%), and TSPE-NN (96.92%). The optimization algorithm also uses the BERT model to explore the input’s features. The introduced method effectively utilizes the BERT model and mask language modeling concept while analyzing the user answers. In addition, the ABC algorithm-based selected weight parameters are utilized here to update the network parameter. The ABC algorithm has a strong exploration ability that helps to predict the appropriate network parameters. In addition, the algorithm can compute the network parameter fastest, robustly, and flexibly. Moreover, the algorithm utilizes a few parameter settings that lead to making the algorithm simple. This process ensures high accuracy and minimum error rate values. Therefore, the BERT incorporated the ABC algorithm to enhance the accuracy of the overall essay score prediction compared to other methods.

Figure 7 (A–C) Efficiency analysis of ABC BERT FTM model.

Discussion

Table 3 illustrates the ABC-BERT-FTM-based AES prediction process in training and testing. The results clearly state that the introduced approach attains 97.087% accuracy, showing that the method correctly classifies the new essay features with minimum deviation error. In addition, the method attains a high F1-score (98.03%) value, which predicts the correctly classified output from the number of collected essay documents. Finally, the system ensures that the high QWK (0.9709) indicates that the system predicts the essay score with an excellent concordance value. The system uses the stochastic gradient descent learning process θt−1−η.∇θJ(θ) for training the features. The learning process computes the model parameter most relevant to computing the output value. In addition, ABC algorithm-based optimized parameters are utilized in this work, reducing the deviation between the output values. Table 3 shows that the ABC-BERT-FTM approach has values from 0.91 to 1 in the QWK guideline table described in Table 2. According to the computation, obtained error values are illustrated in Table 4.

Table 3 Training and testing efficiency.

Model	Training	Testing	
Accuracy	F1-score	QWK	Accuracy	F1-score	QWK	
CA-DL	81.9672	83.3333	0.7813	87.7193	85.4701	0.8398	
CS-VM	75.1880	75.7576	0.7576	75.1880	75.7576	0.7813	
MTL-DL	76.9231	78.1250	0.8333	78.1250	78.1250	0.8333	
HDL	73.5294	74.0741	0.8850	74.0741	74.6269	0.8547	
BERT-FTM	75.644	76.332	0.7532	85.3321	86.3212	0.8432	
ABC-BERT-FTM	96.8538	98.0392	0.9804	97.0874	98.0392	0.9709	

Table 4 Error rate analysis.

Model	Training	Testing	
MSE	RMSE	MSE	RMSE		
CA-DL	0.2247	0.2185	0.2126	0.1605	
CS-VM	0.1966	0.1829	0.1787	0.2070	
MTL-DL	0.1748	0.1710	0.1639	0.1748	
HDL	0.1077	0.1063	0.1049	0.1035	
ABC-BERT-FTM	0.0715	0.0655	0.0605	0.0562	

Table 4 illustrates the error rate analysis of the introduced ABCC-BERT-FTM approach-based essay scoring system on training and testing. Here, it clearly shows that the system ensures the minimum error values (MSE-0.0605 and RMSE-0.0562) compared to the other methods such as CA-DL (MSE-0.212 and RMSE-0.1605), CS-VM (MSE-0.1787 and RMSE-0.2070), MTL-DL (MSE-0.1049 and RMSE-0.1035) and HDL (MSE-0.1049 and RMSE-0.1035).

Conclusions

This article proposes an optimized BERT model-based score prediction process using the ABC algorithm. Initially, the essay was collected from ASAP and ETS datasets, which were investigated using the NLP technique. During this process, irrelevant information like punctuation and URL details are eliminated. Then, lemmatization and stemming are performed to determine the root and essential values of the information. Then, the embedding model is applied to get the words and respective patterns to get the score value. The model uses the multi-attention layer, batch normalization, and SoftMax activation function to predict the output value. Here, network parameters are updated by applying the SGD learning function, which causes the forgetting problem. The problem was reduced by selecting the ABC-based optimized feature. The optimized network parameter improves the score prediction rate by up to 98.51% of accuracy. In the future, feature-engineered systems will be added to the AES task’s multi-head attention by the adjusted BERT architecture to improve contextual information. The present study focuses on the extracted features that consume high computation complexity and feature redundancy. Therefore, effective optimization techniques are required to improve the overall research study in the future. The optimization algorithm selects the features and avoids data redundancy during the feature analysis. In addition, future work should consider cost and time analysis to enhance the research study.

Supplemental Information

Supplemental Information 1 Code used in the analysis.

Additional Information and Declarations

Competing Interests

Author Contributions

Data Availability

The authors declare that they have no competing interests.

Ridha Hussein Chassab conceived and designed the experiments, performed the experiments, analyzed the data, performed the computation work, prepared figures and/or tables, and approved the final draft.

Lailatul Qadri Zakaria conceived and designed the experiments, performed the experiments, analyzed the data, performed the computation work, authored or reviewed drafts of the article, and approved the final draft.

Sabrina Tiun conceived and designed the experiments, performed the experiments, analyzed the data, performed the computation work, authored or reviewed drafts of the article, and approved the final draft.

The following information was supplied regarding data availability:

The ASAP dataset is available at Kaggle: https://www.kaggle.com/c/asap-aes.

The ETS dataset is available at Blanchard, Daniel, et al. ETS Corpus of Non-Native Written English LDC2014T06. Web Download. Philadelphia: Linguistic Data Consortium, 2014. https://doi.org/10.35111/7ez0-x912.

The code is available in the Supplemental File.

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
