# Peer review of "An optimized BERT fine-tuned model using an artificial bee colony algorithm for automatic essay score prediction"

_PeerJ Computer Science, doi:10.7717/peerj-cs.2191_

## Round 0.1 · original submission · Major Revisions

A major criticism for the current version of the paper is the lack of justification for the use of the ABC algorithm as the optimization method for the model. Therefore, you need to add a standard evolutionary algorithm as an additional point of comparison for your ABC algorithm.

The second major criticism is the use of language in the submission. You need to carefully edit your paper to make sure it is readable for a scientific audience.

Please carefully consider all recommendations and comments by the reviewers.

·

Basic reporting

While the overall standard of language (spelling, grammar and so on) is generally sufficient, there are a number of areas where I feel the quality of the writing must be improved. I would recommend another thorough checking of the paper for errors in grammar and spelling. For example, on line 157, the phrase "catastrophic problem" omits the term forgetting. Since this is pivotal to the study it should not be omitted. Lines 190 and 249 also contain some basic language mistakes as examples for the need for greater checking. Finally, I would recommend modifying Figure 3 to include more spacing for the labels as they are too compressed together as well as Figure 4 where errors like "posiitional" and "mulit-head" can be found.

Experimental design

With regards to the experimental design, I have compiled a number of comments that are detailed below.

1. Motivation is required for why the Artificial Bee Colony (ABC) algorithm is chosen to improve the algorithm described in the paper. What is the reason this algorithm in particular was chosen, and is there consideration for what effect a different optimisation algorithm would have on the BERT-FTM process?

2. The ABC algorithm has four parameters that are typically required for optimal performance. The values for these parameters should be listed in the study with explanations for why these values were chosen for the experiments. The choice of parameters could be very relevant to the performance of the ABC-BERT-FTM algorithm and thus it should be mentioned.

3. A description or explanation of the forgetting problem should be placed earlier in the study in order to properly contextualise the research question. This need not be wholly comprehensive but it should be described in an earlier section, and not towards the end of section 3. As this is central to the entire research, it would improve clarity and understanding of why the problem needs to be solved.

4. The datasets chosen for the study provide a relatively large pool of instances. However, a brief motivation for why the datasets were chosen would be a good idea.

Validity of the findings

With regards to the validity of the findings, I have compiled a number of comments that are detailed below.

1. The conclusion of the paper is a good summary of the entire work but the directions for future research are lacking. An brief explanation of why the future research directions should be conducted, with some motivations in the discussion segment, would improve the conclusion.

2. Claims are made about the ACB-BERT-FTM algorithm with regard to its efficiency over other algorithms. While the performance claims are robustly demonstrated, a comparison of the computational costs of the algorithm is omitted. A simple comparison could be to compare the runtimes of the algorithms. While hardware will always be the ultimate determinant, quantifying some of the costs of running the algorithm outside of the number of iterations would further improve the conclusions of the work.

Reviewer 2 ·

Basic reporting

The paper is well structured.

Experimental design

The experimental result analysis should be enhanced.

Validity of the findings

No comment

Additional comments

My main comments are as follows.
1) The motiviation of the work is not very cleary. Please give more analysisi about ABC-BERT-FTM.
2) Why to choose ABC algorihtm as the solver of optimizing BERT model, while there are many state-of-the-art evolutionary algorithms?
3) The analysis of experimental results in the experiment part should be enhanced, to demonstrate the merits of the proposed method.
4) The reference list should be enhanced. Especially, some recent intelligent methods have potential to deal with the problem.
(1)"Enhancing Learning Efficiency of Brain Storm Optimization via Orthogonal Learning Design," IEEE Transactions on Systems, Man, and Cybernetics: Systems, vol. 51, no. 11, pp. 6723-6742, Nov. 2021.)
(2) Learning to Optimize: Reference Vector Reinforcement Learning Adaption to Constrained Many-objective Optimization of Industrial Copper Burdening System, IEEE Transactions on Cybernetics, vol. 52, no. 12, pp. 12698-12711, December 2022.
(3)Dynamic Selection Preference-Assisted Constrained Multiobjective Differential Evolution. IEEE Transactions on Systems, Man, and Cybernetics: Systems, 2021,doi: 10.1109/TSMC.2021.3061698.

---

## Round 0.2 · accepted · Accept

Thanks to the reviewers for their efforts to improve the article. The reviewer and I believe it can be accepted now. Congrats!

·

Basic reporting

No comment.

Experimental design

No comment.

Validity of the findings

No comment.

Additional comments

The quality of the paper has been significantly improved from the first revision. In terms of the comments made during a previous submission, they have been addressed.